# Lazy and Fast Greedy MAP Inference for Determinantal Point Process

**Shinichi Hemmi**
The University of Tokyo, Tokyo, Japan
hemmi.shinichi@gmail.com

**Taihei Oki**
The University of Tokyo, Tokyo, Japan
oki@mist.i.u-tokyo.ac.jp

**Shinsaku Sakaue**
The University of Tokyo, Tokyo, Japan
sakaue@mist.i.u-tokyo.ac.jp

**Kaito Fujii**
National Institute of Informatics, Tokyo, Japan
fujiik@nii.ac.jp

**Satoru Iwata**
The University of Tokyo, Tokyo, Japan
ICReDD, Hokkaido University, Sapporo, Japan
iwata@mist.i.u-tokyo.ac.jp

## Abstract

The maximum a posteriori (MAP) inference for determinantal point processes (DPPs) is crucial for selecting diverse items in many machine learning applications. Although DPP MAP inference is NP-hard, the greedy algorithm often finds high-quality solutions, and many researchers have studied its efficient implementation. One classical and practical method is the lazy greedy algorithm, which is applicable to general submodular function maximization, while a recent fast greedy algorithm based on the Cholesky factorization is more efficient for DPP MAP inference. This paper presents how to combine the ideas of "lazy" and "fast", which have been considered incompatible in the literature. Our lazy and fast greedy algorithm achieves almost the same time complexity as the current best one and runs faster in practice. The idea of "lazy + fast" is extendable to other greedy-type algorithms. We also give a fast version of the double greedy algorithm for unconstrained DPP MAP inference. Experiments validate the effectiveness of our acceleration ideas.

## 1 Introduction

Determinantal point processes (DPPs) offer a popular diversification model in machine learning. Macchi [34] first used DPPs to represent repulsion in quantum physics, and later, DPPs have been used in various scenarios such as recommendation systems [13], document summarization [21, 31], and diverse molecule selection [39]. An important problem in the DPP applications is the *maximum a posteriori* (MAP) inference, which asks to find an item subset with the highest probability. Intuitively, if each item is associated with a vector whose length and direction represent its importance and feature, respectively, then the aim of DPP MAP inference is to select items whose vectors form the largest volume parallelotope, thus selecting important and diverse items.

In DPPs, exact MAP inference is NP-hard [27]. Fortunately, however, the standard greedy algorithm (GREEDY) for submodular function maximization [41] enjoys a $(1 - 1/e)$-approximation guarantee in terms of the log-determinant function value under the monotonicity assumption, and it often finds high-quality solutions in practice. A naive implementation of GREEDY, however, incurs too much computation cost for large instances since evaluating the determinant of a $k \times k$ matrix takes $O(k^\omega)$

36th Conference on Neural Information Processing Systems (NeurIPS 2022).

time, where $\omega \in [2, 3]$ is the matrix-multiplication exponent (usually $\omega = 3$). To overcome this issue, researchers have studied techniques for implementing efficient greedy algorithms. One classical and powerful method is the so-called *lazy* greedy algorithm (LAZYGREEDY) [37], which avoids the redundant computation of function values by making good use of the submodularity. LAZYGREEDY is applicable to submodular function maximization, and it empirically runs much faster than naive GREEDY, although the worst-case running time is not improved. Chen et al. [13] proposed another notable Cholesky-factorization-based method, called the *fast* greedy algorithm (FASTGREEDY). Their algorithm is specialized for DPP MAP inference and provides the fastest $O(knd)$-time implementation of GREEDY for selecting $k$ out of $n$ items represented by $d$-dimensional vectors. Chen et al. [13] also experimentally showed that FASTGREEDY can run faster than LAZYGREEDY.

Since the study of [13], although a pre-processing method for customized DPP MAP inference [23] and fast parallel algorithms for submodular function maximization [3, 4, 18, 8, 17, 14, 29] have been studied, no progress has been made that directly accelerates GREEDY for DPP MAP inference. Since real-world dataset sizes have been growing, a further speed-up of GREEDY is eagerly awaited.

**Our main contribution** is to combine the two ideas, "lazy" and "fast", to develop an even faster implementation of GREEDY for DPP MAP inference, which we call LAZYFASTGREEDY. In the literature, the two ideas have been thought to be incompatible; in fact, experiments in [13] compared FASTGREEDY with LAZYGREEDY without considering their combination. The core idea of "lazy + fast" is widely applicable to other greedy-type algorithms. Below is a summary of our results.

1. We present LAZYFASTGREEDY for cardinality-constrained DPP MAP inference. It takes $O(kn(d+\log n))$ time even in the worst case and is faster than FASTGREEDY in practice. We also extend the idea of "lazy + fast" to other greedy-type algorithms: RANDOMGREEDY [10], STOCHASTICGREEDY [38, 43], and INTERLACEGREEDY [28].

2. We present a "fast" version of DOUBLEGREEDY [11] for unconstrained DPP MAP inference by extending the idea of [13] with Jacobi's complementary minor formula.

3. Experiments on synthetic and real-world datasets validate the empirical effectiveness of our acceleration techniques for the greedy-type algorithms. In particular, our LAZYFAST-GREEDY runs up to about 17 times faster than FASTGREEDY in real-world settings.

Accelerating the greedy variants [10, 38, 28, 43], which enjoy approximation guarantees for non-monotone submodular function maximization, is worthwhile since the log-determinant function is non-monotone in general. Also, note that both cardinality-constrained and unconstrained settings are important in DPP MAP inference and require substantially different technical ideas. Therefore, we study the fast version of [11] separately from the algorithms for the cardinality-constrained setting. In short, we accelerate various important greedy-type algorithms for DPP MAP inference.

## 1.1 Related work

The greedy algorithm (GREEDY) is a popular approach to DPP MAP inference [31]. Han et al. [24] gave a fast but inexact implementation of GREEDY. Later, Chen et al. [13] gave an exact implementation of GREEDY with the same time complexity as that of [24]. Han and Gillenwater [23] studied special cases where kernel matrices of DPPs are generated via customization (or re-weighting) of fixed feature vectors and developed a pre-processing method for accelerating GREEDY. Note that this pre-processing usually takes longer than the algorithm of [13] (see [23, Section 4]), while their algorithm after pre-processing is much faster. In summary, the algorithm of Chen et al. [13] has been the fastest greedy-style algorithm for general DPPs.

A continuous-relaxation-based $1/4$-approximation algorithm for general down-closed constraints was also studied [21]. Our implementation of INTERLACEGREEDY [28] yields a faster $1/4$-approximation algorithm for a special case with a cardinality constraint. Approximation algorithms and inapproximability results of DPP MAP inference (without log) have also been extensively studied [15, 35, 36, 6, 42]. Sampling is another important research subject in DPPs and has been widely studied [2, 33, 16, 22, 1, 12, 32]. Also, we leave extension of our algorithms to non-symmetric DPPs [20], which have recently gained increasing attention, for future work.

Since the log-determinant function is known to be submodular [19], DPP MAP inference has a close connection to submodular function maximization [41]. Besides LAZYGREEDY, STOCHASTIC-GREEDY [38, 43] is a popular fast variant of GREEDY, which we will discuss later. A recent line of

work [3, 4, 18, 8, 17, 14, 29] has studied *adaptive* algorithms for submodular function maximization, where we are allowed to execute polynomially many queries in parallel to reduce the number of sequential rounds. Those studies consider oracle models of submodular functions, whereas we focus on the log-determinant functions and develop fast algorithms without such parallelization.

## 2 Background

Let $[n]$ denote the set $\{1, 2, \ldots, n\}$ for any $n \in \mathbb{N}$. For any $S \subseteq [n]$, $\overline{S}$ denotes its complement $[n] \setminus S$. We use $\mathbf{0}$ and $O$ as all-zero vectors and matrices, respectively, and $\mathbf{1}$ as all-ones vectors (their sizes will be clear from the context). Let $\langle \cdot, \cdot \rangle$ denote the inner product. For a matrix $L \in \mathbb{R}^{n \times n}$ and subsets $S, T \subseteq [n]$, $L[S, T]$ is the submatrix of $L$ indexed by $S$ in rows and $T$ in columns. For brevity, we write $L_{i,j} = L[\{i\}, \{j\}], L[S] = L[S, S], L[S, i] = L[S, \{i\}]$, and $L[i, S] = L[\{i\}, S]$ for any $i, j \in [n]$ and $S \subseteq [n]$. The determinant of $L$ is denoted by $\det L$. Set $\det L[\emptyset] = 1$ by convention. For any $M \in \mathbb{R}^{n \times n}$, we suppose that $M^\top M$ is computed in $O(n^\omega)$ time, which implies that we can compute $\det M$ and $M^{-1}$ (if non-singular) in $O(n^\omega)$ time (see, e.g., [7, Chapter 2]).

**DPP MAP inference.** Let $L \in \mathbb{R}^{n \times n}$ be a positive semi-definite matrix. A probability measure $\mathcal{P}$ on $2^{[n]}$ is called a *determinantal point process* (DPP) with a kernel matrix $L$ if $\mathcal{P}[X = S] \propto \det L[S]$ holds for all $S \subseteq [n]$.[1] MAP inference for DPPs is the problem of finding a subset $S \subseteq [n]$ with the largest $\det L[S]$ value. We suppose that each $i \in [n]$ is associated with a vector $\phi_i \in \mathbb{R}^d$ and that a kernel matrix $L \in \mathbb{R}^{n \times n}$ is given as $L = B^\top B$, where $B = [\phi_1, \phi_2, \ldots, \phi_n] \in \mathbb{R}^{d \times n}$, i.e., $L_{i,j} = \langle \phi_i, \phi_j \rangle$ for $i, j \in [n]$. Note that $\sqrt{\det L[S]}$ represents the volume of the parallelotope spanned by $\{\phi_i \mid i \in S\}$. Hence, if the length and direction of $\phi_i$ indicate $i$'s importance and feature, respectively, the larger value of $\det L[S]$ implies that $S$ contains more important and diverse items. In many situations, we want to select a limited number of items; let $k \in \mathbb{N}$ denote the upper bound. Therefore, MAP inference for DPPs with a cardinality constraint, or $k$-DPP [30], is often considered. Note that since $\mathrm{rank}\, L \leq \min\{n, d\}$, we can assume $k \leq \min\{n, d\}$ without loss of generality.

**Submodular function maximization.** For a set function $f: 2^{[n]} \to \mathbb{R} \cup \{-\infty\}$, the *marginal gain* of $i \in [n]$ with respect to $S \subseteq [n]$ is defined by $f_i(S) = f(S \cup \{i\}) - f(S)$. A set function $f: 2^{[n]} \to \mathbb{R} \cup \{-\infty\}$ is called *monotone* if $f_i(S) \geq 0$ for every $S \subseteq [n]$ and $i \in \overline{S}$. It is called *submodular* if it has the *diminishing returns property*: $f_i(S) \geq f_i(T)$ for every $S \subseteq T \subseteq [n]$ and $i \in \overline{T}$. Since $f(S) = \log \det L[S]$ is submodular, DPP MAP inference can be written as submodular function maximization: $\max_{S \in \mathcal{X}} f(S)$, where $\mathcal{X} \subseteq 2^{[n]}$ is a family of feasible subsets. This paper mostly considers the cardinality-constrained setting, i.e., $\mathcal{X} = \{S \subseteq [n] \mid |S| \leq k\}$ for given $k \in \mathbb{N}$; in Section 4, we study the unconstrained setting, i.e., $\mathcal{X} = 2^{[n]}$. The log-determinant function $f$ is monotone if the smallest eigenvalue of $L$ is at least 1 (see, e.g., [44]), but this is not always the case.

It is well-known that the greedy algorithm (GREEDY) enjoys a $(1 - 1/e)$-approximation guarantee for cardinality-constrained monotone submodular function maximization with $f(\emptyset) \geq 0$ [41]. This approximation ratio is optimal under the evaluation oracle model [40]. GREEDY works as follows: setting $S^{(0)} = \emptyset$, in each $t$th step ($t = 1, \ldots, k$), choose $j_t \in \arg\max\{f_i(S^{(t-1)}) \mid i \in \overline{S^{(t-1)}}\}$ and put $S^{(t)} = S^{(t-1)} \cup \{j_t\}$. We call a solution obtained in this way a *greedy solution*.

Besides GREEDY, many algorithms [11, 10, 28, 43] achieve constant-factor approximations for cardinality-constrained/unconstrained submodular function maximization. These results motivate us to apply submodular-function-maximization algorithms to DPP MAP inference, although constant-factor approximations of the log-determinant value do not imply those of the determinant value.

### 2.1 Lazy greedy algorithm for submodular function maximization

LAZYGREEDY [37] is an efficient implementation of GREEDY for submodular function maximization. As explained above, GREEDY finds $j_t$ by computing marginal gains $f_i(S^{(t-1)})$ for all $i \in \overline{S^{(t-1)}}$. LAZYGREEDY attempts to find $j_t$ more efficiently by keeping an upper bound $\rho_i$ on $f_i(S^{(t-1)})$ for each $i \in \overline{S^{(t-1)}}$, which is an *old* marginal gain, i.e., $\rho_i = f_i(S^{(u_i)})$ for some $u_i \leq t - 1$. In each

---

[1]Strictly speaking, this is the so-called $L$-ensemble DPP, but we here call it a DPP for simplicity.

**Algorithm 1** FASTGREEDY [13] for cardinality-constrained DPP MAP inference

1: $V \leftarrow O$, $d_i^{(0)} \leftarrow \sqrt{L_{i,i}}$ $(\forall i \in [n])$, $S^{(0)} \leftarrow \emptyset$
2: **for** $t = 1$ to $k$ **do**
3:     Take $j_t \in \arg\max_{i \in \overline{S^{(t-1)}}} d_i^{(t-1)}$         ▷ Terminate if $d_{j_t}^{(t-1)} \leq 1$ (i.e., $f_{j_t}(S^{(t-1)}) \leq 0$)
4:     $S^{(t)} \leftarrow S^{(t-1)} \cup \{j_t\}$                   ▷ $S^{(t)} = \{j_1, \ldots, j_t\}$
5:     **for** $i$ in $\overline{S^{(t)}}$ **do**         ▷ Skip Lines 5–7 (updates for the next step) if $t = k$
6:         $V_{i,j_t} \leftarrow (L_{i,j_t} - \langle V[i, S^{(t-1)}], V[j_t, S^{(t-1)}] \rangle)/d_{j_t}^{(t-1)}$   ▷ $V[i, \emptyset] = \mathbf{0}$ for any $i \in [n]$
7:         $d_i^{(t)} \leftarrow \sqrt{\left(d_i^{(t-1)}\right)^2 - V_{i,j_t}^2}$
8: **return** $S^{(k)}$                           ▷ Return $S^{(t-1)}$ if terminates with $t < k$

iteration, LAZYGREEDY picks $i \in \overline{S^{(t-1)}}$ with the largest $\rho_i$ value as the most promising element. It then updates the $\rho_i$ value to the latest marginal gain $f_i(S^{(t-1)})$. If $\rho_i$ is still the largest among $\rho_{i'}$ for all $i' \in \overline{S^{(t-1)}}$, the diminishing returns property guarantees $i \in \arg\max\{f_{i'}(S^{(t-1)}) \mid i' \in \overline{S^{(t-1)}}\}$, and hence it adds $j_t = i$ to $S^{(t-1)}$. LAZYGREEDY thus constructs a greedy solution, deferring updates of upper bounds of unpromising elements. If the upper bounds are managed by a priority queue, every single iteration computes the marginal gain only once and makes $O(\log n)$ comparisons (note that "iteration" is distinguished from "step" of GREEDY). With this contrivance, LAZYGREEDY runs much faster than GREEDY in practice, even though it does not improve the worst-case complexity.

## 2.2 Fast greedy algorithm for DPP MAP inference

We turn to DPP MAP inference with a kernel matrix $L = B^\top B$ and $B \in \mathbb{R}^{d \times n}$. If we apply GREEDY to $f(S) = \log \det L[S]$, it takes $O(k^{\omega+1} n)$ time since computing an $f$ value takes $O(k^\omega)$ time; in addition, computing $L$ at first takes $O(\min\{n^{\omega-1} d, n^2 d^{\omega-2}\})$ time. FASTGREEDY [13] provides an $O(knd)$-time implementation of GREEDY for cardinality-constrained DPP MAP inference.

Algorithm 1 describes the procedure of FASTGREEDY, which is based on the Cholesky factorization of $L$ with maximum pivoting. The Cholesky decomposition produces a matrix $V \in \mathbb{R}^{n \times n}$ such that $L = VV^\top$ and $PV$ is lower triangular for some permutation matrix $P$. We call $V$ a *Cholesky factor* of $L$. In each $t$th step, Lines 5–7 calculate the $t$th column of $PV$ via backward substitution (see Fig. 1a). The following Proposition 2.1 implies an important fact that once $V[i, S^{(t)}]$ is filled, we can obtain $f_i(S^{(t)})$ from $d_i^{(t)}$ computed in Line 7, which is equal to the diagonal $V_{i,i}$ of the current Cholesky factor. Although the proposition is already proved in [13], we present a proof sketch since it would be helpful to understand the subsequent discussion (see [13] for the complete proof).

**Proposition 2.1** ([13]). *For $t = 0, 1, \ldots, k-1$, it holds that $f_i(S^{(t)}) = 2 \log d_i^{(t)}$ for every $i \in \overline{S^{(t)}}$.*

*Proof sketch.* The proof is by induction on $t$. If $t = 0$, it holds that $f_i(S^{(t)}) = f(\{i\}) - f(\emptyset) = \log L_{i,i} = 2 \log d_i^{(0)}$ for $i \in [n]$. Given a Cholesky factor $V[S^{(t)}]$ of $L[S^{(t)}]$, for $i \in \overline{S^{(t)}}$, we have

$$L[S^{(t)} \cup \{i\}] = \begin{bmatrix} L[S^{(t)}] & L[S^{(t)}, i] \\ L[i, S^{(t)}] & L_{i,i} \end{bmatrix} = \begin{bmatrix} V[S^{(t)}] & \mathbf{0} \\ V[i, S^{(t)}] & d_i^{(t)} \end{bmatrix} \begin{bmatrix} V[S^{(t)}]^\top & V[i, S^{(t)}]^\top \\ \mathbf{0}^\top & d_i^{(t)} \end{bmatrix}.$$

Hence, we have $\log \det L[S^{(t)} \cup \{i\}] = \log \left(d_i^{(t)} \det V[S^{(t)}]\right)^2 = 2 \log d_i^{(t)} + \log \det L[S^{(t)}]$, which implies $\log \det L[S^{(t)} \cup \{i\}] - \log \det L[S^{(t)}] = 2 \log d_i^{(t)}$. Thus, the statement holds. $\square$

Therefore, by iteratively adding $j_t \in \arg\max_{i \in \overline{S^{(t-1)}}} d_i^{(t-1)}$ to $S^{(t-1)}$ as in Algorithm 1, we can obtain a greedy solution. Since $L_{i,j_t} = \langle \phi_i, \phi_{j_t} \rangle$ is computed in $O(d)$ time and $d \geq k$, Line 6 takes $O(d)$ time. This is repeated $O(kn)$ times, hence the total time complexity of $O(knd)$.

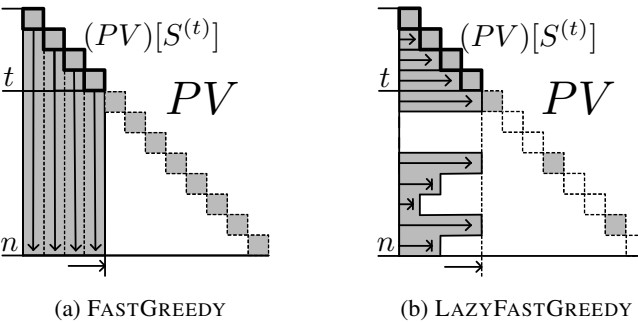

(a) FASTGREEDY  (b) LAZYFASTGREEDY

Figure 1: Images of Cholesky factors computed by FASTGREEDY and LAZYFASTGREEDY. Off-diagonals are shaded if they are already computed, and arrows represent the direction in which the computation of off-diagonals proceeds. Diagonals are shaded if they give the latest marginal gains, each of which becomes available once $|S^{(t)}|$ off-diagonals in the same row are computed (shaded). Diagonals with bold lines correspond to elements that are already selected.

## 3 Lazy and fast algorithms for cardinality-constrained DPP MAP inference

### 3.1 Lazy and fast greedy algorithm

We combine LAZYGREEDY and FASTGREEDY to obtain even faster LAZYFASTGREEDY.

As described above, LAZYGREEDY is designed for submodular function maximization under the oracle model, where a marginal gain is computed for each $i \in \overline{S^{(t-1)}}$. Meanwhile, the core idea of FASTGREEDY is to obtain marginal gains of all $i \in \overline{S^{(t-1)}}$ efficiently by computing a new column of a Cholesky factor, as in Fig. 1a. To combine these seemingly incompatible methods, we need to take a closer look at how FASTGREEDY updates the entries in $V$. The following observation is obvious but elucidates the essential row-wise independence of the updates of Cholesky-factor entries.

**Observation 3.1.** *In each $t$th step in Algorithm 1, for each $i \in \overline{S^{(t)}}$, $V_{i,j_t}$ in Line 6 is computed from $d_{j_t}^{(t-1)}$, $L_{i,j_t}$, and $V[\{i, j_t\}, S^{(t-1)}]$. Thus, conditioned on $S^{(t)}$, the $i$th row of $V[\overline{S^{(t)}}, S^{(t)}]$ can be updated independently for each $i \in \overline{S^{(t)}}$ in a sense that computing $V_{i,j_t}$ only requires $V_{i,j_{t'}}$ for $t' < t$ and $V[j_t, S^{(t-1)}]$, which is included in the already fixed Cholesky factor $V[S^{(t)}]$.*

That is, in Fig. 1b, once $(PV)[S^{(t)}]$ is fixed, we can update the $i$th row independently for each $i \in \overline{S^{(t)}}$. In the words of submodular function maximization, we can compute $f_i(S^{(t)})$ if $f_{j_{t'}}(S^{(t'-1)})$ and $f_i(S^{(t'-1)})$ for $t' \le t$ are available. This enables us to apply the idea of lazy updates to FASTGREEDY.

Algorithm 2 describes our LAZYFASTGREEDY, and Fig. 1b illustrates how entries in $V$ are updated. In each iteration, it picks the most promising element $i$ in Line 3, as with LAZYGREEDY. Then, it calls UPDATEROW to compute the entries of $V[i, S]$ via backward substitution. Once $V[i, S]$ is filled, the resulting $d_i$ computed in Line 12 satisfies $2 \log d_i = f_i(S)$ by Proposition 2.1, thus obtaining the latest marginal gain of $i$. Lines 5–7 check whether the latest $f_i(S)$ is still the largest among the (old) marginal gains of $i' \in \overline{S}$; if so, $j_{|S|+1} = i$ is added to $S$. Note that the deferred updates of $d_{i'}$ ($i' \in \overline{S} \setminus \{i\}$) do not matter when deciding whether to add $i$ to $S$ due to the submodularity. Here, $\boldsymbol{d} = (d_1, \ldots, d_n)$ plays the role of upper bounds $(\rho_1, \ldots, \rho_n)$ of LAZYGREEDY and is maintained by a priority queue. LAZYFASTGREEDY thus finds a greedy solution by exactly mimicking the behavior of LAZYGREEDY while computing marginal gains efficiently as with FASTGREEDY.

The vector $\boldsymbol{u} = (u_1, \ldots, u_n)$ keeps track of when the upper bounds are last updated; specifically, each $u_i \in \{0, 1, \ldots, k-1\}$ indicates that the upper bound $d_i$ is last updated with respect to $\{j_1, \ldots, j_{u_i}\}$, i.e., $2 \log d_i = f_i(\{j_1, \ldots, j_{u_i}\})$. Since UPDATEROW starts to fill $V[i, S]$ from the $(u_i + 1)$st entry, no off-diagonals $V_{i,j_t}$ are computed more than once in Line 11. Thus, the complexity of Algorithm 2 depends on the number of computed off-diagonals. We denote it by $u = \sum_{i \in [n]} u_i$ with $u_i$ values at the end of Algorithm 2. The $u$ value changes in the range of $[k(k-1)/2, (k-1)(n-k/2)]$ depending on how well the lazy update works and affects the overall time complexity as follows.

---

**Algorithm 2** LAZYFASTGREEDY for cardinality-constrained DPP MAP inference

---

1: $V \leftarrow O, \boldsymbol{d} \leftarrow \left(\sqrt{L_{i,i}}\right)_{i \in [n]}, \boldsymbol{u} \leftarrow \boldsymbol{0}, S \leftarrow \emptyset$      $\triangleright \boldsymbol{d}$ is maintained by a priority queue
2: **while** $|S| < k$ **do**
3:      Take $i \in \arg\max_{i' \in \overline{S}} d_{i'}$      $\triangleright$ Terminate if $d_i \leq 1$ (i.e., $f_i(S) \leq 0$)
4:      UPDATEROW$(V, \boldsymbol{d}, \boldsymbol{u}; i, S, L)$      $\triangleright$ Nothing is done if $|S| = 0$
5:      **if** $d_i \geq \max_{i' \in \overline{S}} d_{i'}$ **then**      $\triangleright$ Otherwise insert $d_i$ into the priority queue
6:          $j_{|S|+1} \leftarrow i$
7:          $S \leftarrow S \cup \{j_{|S|+1}\}$
8: **return** $S$

9: **function** UPDATEROW$(V, \boldsymbol{d}, \boldsymbol{u}; i, S, L)$      $\triangleright S = \{j_1, j_2, \ldots, j_{|S|}\}$
10:      **for** $t = u_i + 1, u_i + 2, \ldots, |S|$ **do**
11:          $V_{i,j_t} \leftarrow (L_{i,j_t} - \langle V[i, S^{(t-1)}], V[j_t, S^{(t-1)}] \rangle)/d_{j_t}$      $\triangleright S^{(t-1)} = \{j_1, \ldots, j_{t-1}\}$
12:          $d_i \leftarrow \sqrt{d_i^2 - V_{i,j_t}^2}$
13:      $u_i \leftarrow |S|$      $\triangleright$ This line is not needed in Algorithm 3

---

**Theorem 3.2.** *Algorithm 2 returns a greedy solution in* $\mathrm{O}(nd + u(d + \log n))$ *time. If the lazy update works best and worst, it runs in* $\mathrm{O}((n + k^2)d)$ *and* $\mathrm{O}(kn(d + \log n))$ *time, respectively.*

*Proof.* Algorithm 2 returns a greedy solution as explained above. We below discuss the running time.

At the beginning, we need $\mathrm{O}(nd)$ time to compute $L_{i,i} = \langle \boldsymbol{\phi}_i, \boldsymbol{\phi}_i \rangle$ for $i = 1, \ldots, n$. In UPDATEROW, an access to $L_{i,j_t}$ in Line 11 takes $\mathrm{O}(d)$ time, and the inner product takes $\mathrm{O}(k)$ ($\lesssim \mathrm{O}(d)$) time. This computation is done $u$ times, and thus the total computation time caused by UPDATEROW is $\mathrm{O}(ud)$. In Line 5, we need $\mathrm{O}(\log n)$ time to update the priority queue if $d_i < \max_{i' \in \overline{S}} d_{i'}$, which can hold only when at least one off-diagonal is computed in UPDATEROW. Therefore, Line 5 takes $\mathrm{O}(u \log n)$ time in total. Thus, the overall time complexity is $\mathrm{O}(nd + u(d + \log n))$.

Let $S$ be the output of Algorithm 2 and $P$ a permutation matrix such that $(PV)[S]$ is lower triangular. In the best case, UPDATEROW is called up to $k$ times and $u = k(k-1)/2$ off-diagonals of $(PV)[S]$ are computed. Moreover, updates of the priority queue are done only up to $k$ times, taking $\mathrm{O}(k \log n)$ ($\lesssim \mathrm{O}(nd)$) time in total. Thus, it runs in $\mathrm{O}((n + k^2)d)$ time. In the worst case, Algorithm 2 calculates the off-diagonals of $(PV)[S]$ and all the entries of $V[\overline{S}, S \setminus \{j_k\}]$; the total number of those entries is $u = k(k-1)/2 + (k-1)(n-k) = (k-1)(n-k/2)$. Hence, it takes $\mathrm{O}(kn(d + \log n))$ time. $\quad\square$

The best-case time complexity is better than $\mathrm{O}(knd)$ of FASTGREEDY if $k = \mathrm{o}(n)$. Even in the worse case, it is as fast as FASTGREEDY if $d = \Omega(\log n)$. Note that both $k = \mathrm{o}(n)$ and $d = \Omega(\log n)$ are true in most practical situations. Experiments in Section 5 demonstrate that LAZYFASTGREEDY can run much faster than FASTGREEDY in practice.

### 3.2 Extension to random, stochastic, and interlace greedy algorithms

The core idea of LAZYFASTGREEDY can be used for speeding up other greedy-type algorithms: RANDOMGREEDY [10], STOCHASTICGREEDY [38, 43], and INTERLACEGREEDY [28], which enjoy $1/e$-, $1/4$-, and $1/4$-approximation guarantees, respectively, for non-monotone submodular function maximization with a cardinality constraint. Note that the guarantees for the non-monotone case are essential in DPP MAP inference since the log-determinant function is non-monotone in general. Due to the space limitation, we present the details of those extensions in Appendix A.

## 4 Fast double greedy algorithm for unconstrained DPP MAP inference

This section discusses unconstrained DPP MAP inference with a kernel matrix $L = B^\top B$, where $B \in \mathbb{R}^{d \times n}$. In this setting, if $f(S) = \log \det L[S]$ is monotone, $S = [n]$ is a trivial optimal solution. Thus, we suppose $f$ to be non-monotone. We also assume $L$ to be positive definite since the algorithm of [11] discussed below requires $f(S) > -\infty$ for any $S \subseteq [n]$. Note that this implies $d \geq n$.

---

**Algorithm 3** FASTDOUBLEGREEDY for unconstrained DPP MAP inference

---

1: Compute $L = B^\top B$ and $L^{-1}$
2: $V \leftarrow O, W \leftarrow O, \boldsymbol{d} \leftarrow \left(\sqrt{L_{i,i}}\right)_{i \in [n]}, \boldsymbol{e} \leftarrow \left(\sqrt{(L^{-1})_{i,i}}\right)_{i \in [n]}, S \leftarrow \emptyset$
3: **for** $i = 1$ to $n$ **do**
4:      UPDATEROW$(V, \boldsymbol{d}, \boldsymbol{0}; i, S, L)$              $\triangleright$ $S$ is sorted in order of $1, 2, \ldots, n$
5:      UPDATEROW$(W, \boldsymbol{e}, \boldsymbol{0}; i, [i] \setminus S, L^{-1})$       $\triangleright$ $[i] \setminus S$ is sorted in order of $1, 2, \ldots, n$
6:      $a_i \leftarrow \max\{2 \log d_i, 0\}, b_i \leftarrow \max\{2 \log e_i, 0\}$
7:      $S \leftarrow S \cup \{i\}$ w.p. $a_i/(a_i + b_i)$         $\triangleright$ Implicity update $T = \overline{[n] \setminus S}$ w.p. $b_i/(a_i + b_i)$
8: **return** $S$

---

A famous algorithm for unconstrained submodular function maximization is DOUBLEGREEDY [11], a randomized $1/2$-approximation algorithm. Although it calls an evaluation oracle only O$(n)$ times, its naive implementation is too costly for large DPP MAP inference instances since computing the log-determinant function value takes O$(n^\omega)$ time, which will lead to the total time of O$(n^{\omega-1}d + n^{\omega+1})$. We below extend the idea of FASTGREEDY [13] to DOUBLEGREEDY and obtain its O$(n^{\omega-1}d + n^3)$-time implementation for unconstrained DPP MAP inference.

DOUBLEGREEDY maintains two subsets $S$ and $T$, which are initially set to $S = \emptyset$ and $T = [n]$. For $i = 1, \ldots, n$, it computes $a_i = \max\{f_i(S), 0\}$ and $b_i = \max\{-f_i(T \setminus \{i\}), 0\}$, and then either adds $i$ to $S$ with probability $a_i/(a_i + b_i)$ or removes $i$ from $T$ with probability $b_i/(a_i + b_i)$.[2] Note that $T = [n] \setminus ([i] \setminus S) = \overline{[i] \setminus S}$ always holds. Finally, it returns $S$ (or equivalently $T = \overline{[n] \setminus S} = S$).

As for the growing subset $S$, we can efficiently compute marginal gains $f_i(S)$ by incrementally updating a Cholesky factor, as with FASTGREEDY. When it comes to the shrinking subset $T$, however, we cannot directly use the efficient incremental update for computing $-f_i(T \setminus \{i\})$. If we naively compute it in each step, it takes O$(n^\omega)$ time, resulting in the same time complexity as the naive implementation. Our key idea for overcoming this difficulty is to use the following Jacobi's complementary minor formula (see, e.g., [9]).

**Proposition 4.1.** *Let $L \in \mathbb{R}^{n \times n}$ be a non-singular matrix and $I, J \subseteq [n]$ be subsets with $|I| = |J|$. Then, it holds that $\det L[I, J] = (-1)^{\sum_{i \in I} i + \sum_{j \in J} j} \det L \det L^{-1}[\overline{I}, \overline{J}]$.*

This formula provides a lemma that enables us to compute $-f_i(T \setminus \{i\})$ via incremental updates.

**Lemma 4.2.** *Let $L \in \mathbb{R}^{n \times n}$ be positive definite. For any $S \subseteq [n]$, define $f(S) = \log \det L[S]$ and $g(S) = \log \det L^{-1}[S]$. Then, $g(S \cup \{i\}) - g(S) = f(\overline{S} \setminus \{i\}) - f(\overline{S})$ holds for any $S \subseteq [n]$ and $i \in [n]$.*

*Proof.* By using Proposition 4.1, we can prove the claim as follows:
$$
\begin{aligned}
f(\overline{S} \setminus \{i\}) - f(\overline{S}) &= \log \det L[\overline{S} \setminus \{i\}] - \log \det L[\overline{S}] \\
&= \log \det L \det L^{-1}[[n] \setminus (\overline{S} \setminus \{i\})] - \log \det L \det L^{-1}[[n] \setminus \overline{S}] \\
&= \log \det L^{-1}[S \cup \{i\}] - \log \det L^{-1}[S] \\
&= g(S \cup \{i\}) - g(S). \qquad \square
\end{aligned}
$$

Lemma 4.2 implies $g(([i] \setminus S) \cup \{i\}) - g([i] \setminus S) = f(\overline{([i] \setminus S)} \setminus \{i\}) - f(\overline{[i] \setminus S}) = -f_i(T)$. Therefore, by computing $L = B^\top B$ and $L^{-1}$ in O$(n^{\omega-1}d)$ ($\gtrsim$ O$(n^\omega)$) time in advance, we can compute $-f_i(T \setminus \{i\})$ in each step by incrementally updating a Cholesky factor of size $|[i] \setminus S|$.

Algorithm 3 presents our FASTDOUBLEGREEDY based on this idea. In each $i$th step, the first and second calls to UPDATEROW, defined in Algorithm 2, fill $V[i, S]$ and $W[i, [i] \setminus S]$, respectively. Hence, Proposition 2.1 and Lemma 4.2 imply $2 \log d_i = f_i(S)$ and $2 \log e_i = g_i([i] \setminus S) = -f_i(T)$. Thus, Algorithm 3 exactly mimics the behavior of DOUBLEGREEDY while efficiently computing marginal gains via incremental updates of Cholesky factors. In UPDATEROW defined in Algorithm 2, since $L$ and $L^{-1}$ are already computed, each $V_{i,j_t}$ is calculated in O$(n)$ time; therefore, a single call to UPDATEROW takes O$(n^2)$ time. Since UPDATEROW is called $2n$ times, Lines 2–7 construct a solution in O$(n^3)$ time. In total, Algorithm 3 runs in O$(n^{\omega-1}d + n^3)$ time.

---

[2] The algorithm thus sequentially examines all elements, and hence there is no room for the lazy update.

# 5 Experiments

We evaluate the effectiveness of our acceleration techniques on synthetic and real-world datasets. Section 5.1 examines speed-ups of GREEDY for cardinality-constrained DPP MAP inference, and Section 5.2 focuses on DOUBLEGREEDY for the unconstrained setting. We present experimental results on RANDOMGREEDY, STOCHASTICGREEDY, and INTERLACEGREEDY in Appendix A.4.

The algorithms are implemented in C++ with library Eigen 3.4.0 for matrix computations. Experiments are conducted using a compiler GCC 10.2.0 on a computer with $3.8\,\mathrm{GHz}$ Intel Xeon Gold CPU and $800\,\mathrm{GB}$ RAM.

We use synthetic and two real-world datasets, Netflix Prize [5] and MovieLens [25]. Each dataset provides a matrix $B \in \mathbb{R}^{d \times n}$ consisting of column vectors $\phi_1, \phi_2, \ldots, \phi_n \in \mathbb{R}^d$ of $n$ items, which defines an $n \times n$ kernel matrix $L = B^\top B$. Below we explain the item vectors of each dataset.

**Synthetic datasets.** We use the setting of [21]. Each entry of $\phi_i \in \mathbb{R}^d$ is independently drawn from the standard normal distribution, $\phi_{ij} \sim \mathcal{N}(0, 1)$. As a result, the kernel matrix $L$ conforms to a Wishart distribution with $n$ degrees of freedom and an identity covariance matrix, i.e., $L \sim \mathcal{W}(I, n, n)$. We consider various $n$ values in the experiments below, and we always set the vector length $d$ to $n$.

**Real-world datasets.** Both Netflix Prize and MovieLens datasets contain users' ratings of movies from one to five, where we regard a movie as an item. Following [13], we binarize the ratings based on whether it is greater than or equal to four. After that, we eliminate movies that result in all-zero vectors and users who result in all-zero ratings since those are redundant. Consequently, the Netflix Prize dataset has $n = 17770$ movies and $d = 478615$ users with $56919190$ ratings; the MovieLens dataset has $n = 40858$ movies and $d = 162342$ users with $12452811$ ratings.

**$B$- and $L$-input settings.** It is important to care about whether $L = B^\top B$ is computed in advance or not. In practice, a matrix $B \in \mathbb{R}^{d \times n}$ of item vectors is often given. Then, computing $L = B^\top B$ in advance takes $\mathrm{O}(\min\{n^{\omega-1}d, n^2 d^{\omega-2}\})$ time, which we should avoid when $n$ is large since the running time of LAZYFASTGREEDY (and FASTGREEDY) increases only linearly in $n$. On the other hand, we are sometimes given a pre-computed kernel matrix $L$, and we can access $L_{i,j}$ in $\mathrm{O}(1)$ time.

We below consider both settings, called $B$- and $L$-input settings, respectively. In the $L$-input setting, we exclude the time to compute $L = B^\top B$ from consideration. Under this condition, FASTGREEDY takes $\mathrm{O}(k^2 n)$ time and LAZYFASTGREEDY does $\mathrm{O}(n + u(k + \log n))$, where the first $\mathrm{O}(n)$ term is for constructing a priority queue. By similar reasoning to that in the proof of Theorem 3.2, it runs in $\mathrm{O}(n + k^3 + k \log n)$ and $\mathrm{O}(kn(k + \log n))$ time if the lazy update works best and worst, respectively.

## 5.1 Greedy algorithm for cardinality-constrained DPP MAP inference

We compare the running time of GREEDY, LAZYGREEDY, FASTGREEDY, and LAZYFASTGREEDY, which we here call Naive, Lazy, Fast, and LazyFast, respectively, for short. As regards synthetic datasets, we consider two settings that fix either $n$ or $k$: (i) $n = 6000$ and $k = 1, 2, \ldots, n$, and (ii) $k = 200$ and $n = 1000, 2000, \ldots, 10000$. We set the timeout periods of (i) and (ii) to 3600 and 60 seconds, respectively. Regarding real-world datasets, $n$ is fixed as explained above and we increase $k = 1, 2, \ldots, n$. We set the timeout period to 3600 seconds. With the Netflix (MovieLens) dataset, the objective value peaked with $k = 17762$ ($k = 18763$), and thus we stopped increasing $k$ after that.

Figures 2 and 3 present the results on synthetic and real-world datasets, respectively. The curves in the runtime figures represent that faster algorithms can return greedy solutions to instances with larger $k$ or $n$ values within the timeout periods. The rightmost figures present the number of off-diagonals of Cholesky factors computed by Fast and LazyFast. As explained in Section 3.1, while Fast always computes all the off-diagonals of $V[[n], S]$, LazyFast does not due to the lazy update.

LazyFast was the fastest in all the settings. In particular, in the synthetic (ii) and real-world settings, LazyFast computed fewer off-diagonals than Fast, thus running the fastest. In the synthetic setting (i), although LazyFast computed almost all off-diagonals in $V[[n], S]$, it was still faster than Fast. For example, for the $L$-input setting with $n = k = 6000$, Fast and LazyFast took $37.4$ and $13.7$ seconds, respectively. This unexpected speed-up is caused by the cache efficiency of LazyFast. Specifically, every call to UPDATEROW computes off-diagonals from $V_{i,j_{u_i+1}}$ to $V_{i,j_{|S|}}$ by accessing entries only in

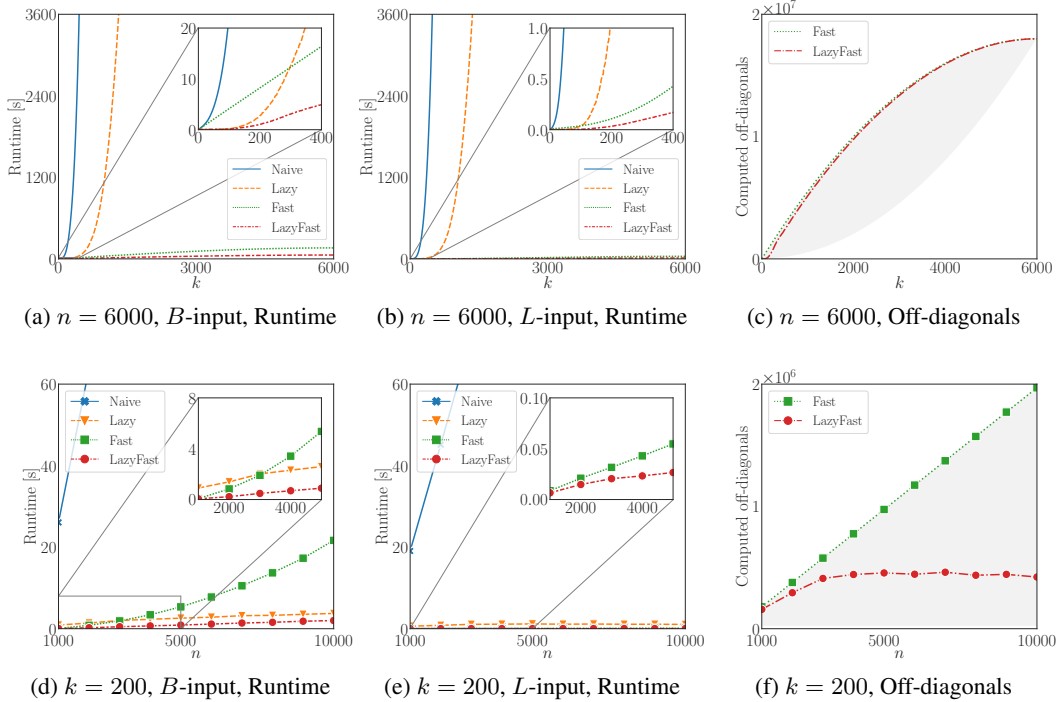

(a) $n = 6000$, $B$-input, Runtime  (b) $n = 6000$, $L$-input, Runtime  (c) $n = 6000$, Off-diagonals

(d) $k = 200$, $B$-input, Runtime  (e) $k = 200$, $L$-input, Runtime  (f) $k = 200$, Off-diagonals

Figure 2: Results on synthetic datasets. In the four runtime figures, enlarged views of lower left parts are shown for visibility. In Figs. 2c and 2f, the gray band indicates the range of the possible number of computed off-diagonals: $[k(k-1)/2, (k-1)(n-k/2)]$.

$V[\{i, j_{u_i+1}, \ldots, j_{|S|}\}, S^{(|S|-1)}]$. This process virtually creates blocks in a Cholesky factor, enabling the cache-efficient computation of off-diagonals via blocking (see, e.g., [26, Section 2.3]). In fact, the cache miss rates of Fast and LazyFast for the above example were 71.3% and 3.3%, respectively.

Another interesting observation in the synthetic (ii) and real-world settings is that while Fast was often faster than Lazy in the $L$-input setting, the opposite occurred in the $B$-input setting. This is because computing an off-diagonal in the $B$-input setting is costly relative to the $L$-input setting. As a result, avoiding the redundant computation of off-diagonals by the lazy update tends to be more effective than computing marginal gains efficiently via the Cholesky factorization.

## 5.2 Double greedy algorithm for unconstrained DPP MAP inference

We compare naive DOUBLEGREEDY and our FASTDOUBLEGREEDY by applying them to unconstrained DPP MAP inference on three datasets: synthetic ($n = 6000$), Netflix Prize, and MovieLens. Since kernel matrices $L$ in the real-world datasets are singular, we use kernel matrices computed as $L = 0.9B^\top B + 0.1I$ in this section, ensuring that the resulting matrices $L$ are positive definite. In this section, we set the timeout period to one day (86400 seconds).

Table 1 presents the results. Note that both algorithms require $L$ to be computed in advance. Therefore, we measured the time of computing $B^\top B$ (Product) separately from the time of constructing solutions (Greedy). Our FASTDOUBLEGREEDY additionally requires $L^{-1}$ to be computed in advance for accelerating the solution construction; therefore, we also measured the time of computing $L^{-1}$ (Inverse) separately. As in Table 1, naive DOUBLEGREEDY took so long for solution construction that it failed to return solutions to real-world instances in one day. By contrast, our FASTDOUBLEGREEDY constructed solutions far faster and succeeded in returning solutions to all the instances.

Also, the computation of $B^\top B$ (Product) took a considerably long time relative to the running time of LAZYFASTGREEDY in the previous section. Therefore, as mentioned above, when a matrix $B$ of item vectors is given and our goal is to obtain a greedy solution for small $k = o(n)$, we should avoid computing the kernel matrix $L = B^\top B$ in advance.

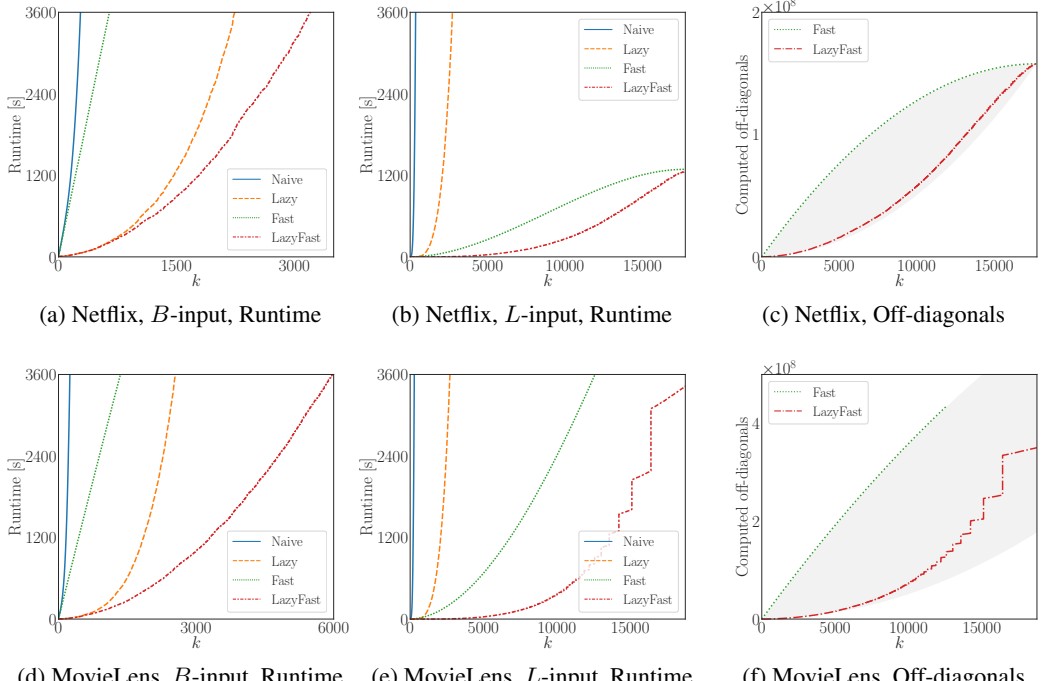

(a) Netflix, $B$-input, Runtime      (b) Netflix, $L$-input, Runtime      (c) Netflix, Off-diagonals

(d) MovieLens, $B$-input, Runtime    (e) MovieLens, $L$-input, Runtime    (f) MovieLens, Off-diagonals

Figure 3: Results on real-world datasets. In Figs. 3c and 3f, the gray band indicates the range of the possible number of computed off-diagonals: $[k(k-1)/2, (k-1)(n-k/2)]$.

Table 1: Running time [s] of DOUBLEGREEDY and FASTDOUBLEGREEDY

| Dataset | DOUBLEGREEDY | | | FASTDOUBLEGREEDY | | | |
|---|---|---|---|---|---|---|---|
| | Product | Greedy | **Total** | Product | Inverse | Greedy | **Total** |
| Synthetic | | | | | | | |
| $n = 2000$ | 1.1 | 258.6 | 259.7 | 1.1 | 1.8 | 0.7 | 3.6 |
| $n = 4000$ | 8.7 | 5422.4 | 5431.1 | 8.7 | 16.9 | 10.0 | 35.6 |
| $n = 6000$ | 30.0 | 36233.1 | 36263.1 | 30.0 | 59.8 | 34.6 | 124.4 |
| $n = 8000$ | 70.2 | > 86400.0 | — | 70.2 | 151.2 | 103.1 | 324.5 |
| $n = 10000$ | 137.9 | > 86400.0 | — | 137.9 | 294.2 | 182.6 | 614.7 |
| Netflix Prize | 20561.6 | > 86400.0 | — | 20561.6 | 1706.7 | 916.2 | 23184.5 |
| MovieLens | 37829.5 | > 86400.0 | — | 37829.5 | 21999.2 | 6337.3 | 66166.0 |

# Acknowledgements

The authors thank anonymous reviewers for their valuable comments. This work was supported by JST ERATO Grant Number JPMJER1903 and JSPS KAKENHI Grant Number JP22K17853.

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
