# OpenReview forum: "Lazy and Fast Greedy MAP Inference for Determinantal Point Process"
_NeurIPS.cc/2022/Conference — NeurIPS 2022 Accept_

### Official Review · Reviewer_LKp4 · 2022-07-07

**Rating:** 6
**Confidence:** 3
**Soundness:** 3 good
**Presentation:** 3 good
**Contribution:** 3 good

**Summary:**

This paper considers the problem of efficiently computing an approximation to the maximum a posteriori inference (MAP) of a discrete determinantal point process (DPP). Specifically, $L$-ensemble DPPs are considered, i.e., DPPs for which the likelihood of a point pattern has a simple formula using a likelihood matrix kernel $L\succeq 0$.

This work combines two existing approaches: the lazy greedy algorithm for submodular function maximization and the fast greedy algorithm for DPP MAP inference. The latter relies on  Cholesky factorization updates of the likelihood kernel.

Two cases are discussed: cardinality-constrained and unconstrained DPP MAP inference.

Numerical simulations illustrate the interest of this approach.

**Questions:**

Here are some comments and suggestions.

- At line 87, I think it is worth mentioning that $L$ is the likelihood kernel of an $L$-ensemble DPP. Basically, the authors do not work with any DPP but they rather deal with a subclass of DPPs for which the correlation kernel has the form $L(L+\mathbb{I})^{-1}$ and which are called $L$-ensemble DPPs. I suggest that the authors briefly mention this fact somewhere by possibly pointing to an extra paragraph in Supplementary Material.


- At line 105, it is mentioned that the log-determinant function $f$ is monotone if the smallest eigenvalue of $L$ is at least $1$.
Could you give a pointer to a reference? For instance, Sharma, Deshpande, Kapoor ICML 2015.

- At line 123, there is a typo: the authors write *hence it adds $j_t = i$ to $\overline{S^{(t-1)}}$*.
 It should be *it adds $j_t = i$ to $S^{(t-1)}$*.

- At line 240, there is a missing $\det$. The correct expression should be $g(S) = \log \det L^{-1}[S]$.

Minor comments:

- Line 15: the use of the singular is strange. I suggest 'DPPs have been...'

- Line 70: *to have the submodularity* &larr; to be submodular?

- line 187: The notation $U$ (uppercase) is a bit misleading since $U$ is a real number and not a matrix (or a set). It is maybe better to use a lowercase letter.

**Limitations:**

The authors discuss the limitation of their results and summarize them in the checklist. I do not expect negative societal impacts from this algorithmic study.

**Strengths And Weaknesses:**

originality: To my knowledge, this idea has not yet been discussed elsewhere.

quality: This paper is carefully written and understandable. It contains very few typos. Although it is not a big issue, there is no conclusion section.

clarity: I found the text and the maths clearly written. A few suggestions of improvements are given below.

significance: The results of this paper are probably useful in practice. On the other hand, this work combines two existing techniques and this combination may be appear as a weak contribution. However, since this combination is not straightforward,  this paper is interesting for the ML/DPP community.

---

> ### Author Response · Authors · 2022-08-02
> **Response to Reviewer LKp4**
>
> We appreciate the reviewer's careful reading and many constructive comments.
>
> &nbsp;
> > At line 87, I think it is worth mentioning that $L$ is the likelihood kernel of an $L$-ensemble DPP. Basically, the authors do not work with any DPP but they rather deal with a subclass of DPPs for which the correlation kernel has the form $L(L+\mathbb{I})^{-1}$ and which are called $L$-ensemble DPPs. I suggest that the authors briefly mention this fact somewhere by possibly pointing to an extra paragraph in Supplementary Material.
>
> Thank you for the insightful suggestion. In the final version, we will mention it in the supplementary (or in the main body using an additional page).
>
> &nbsp;
> > At line 105, it is mentioned that the log-determinant function $f$ is monotone if the smallest eigenvalue of $L$ is at least 1. Could you give a pointer to a reference? For instance, Sharma, Deshpande, Kapoor ICML 2015.
>
> Thank you for the suggestion and for providing the reference. We will present a pointer to the reference.
>
> &nbsp;
> > At line 123, there is a typo: the authors write hence it adds $j_t=i$ to $\overline{S^{(t-1)}}$. It should be it adds $j_t=i$ to $S^{(t-1)}$.
> >
> > At line 240, there is a missing $\det$. The correct expression should be $g(S)=\log\det L^{-1}[S]$.
>
> Thank you for pointing them out. We will fix the typos in the camera-ready version.

---

> > ### Comment · Reviewer_LKp4 · 2022-08-06
> > **Comment**
> >
> > I thank the authors for commenting about my remarks.

---

### Official Review · Reviewer_f4YC · 2022-07-10

**Rating:** 4
**Confidence:** 5
**Soundness:** 3 good
**Presentation:** 4 excellent
**Contribution:** 1 poor

**Summary:**

This paper studies a fast algorithm for greedy MAP inference of Determinantal Point Processes (DPPs). The authors first apply a lazy evaluation trick (widely used in submodular maximization) to the Cholesky-factorization-based greedy DPP MAP algorithm (FastGreedy). They analyze the runtime of the proposed algorithm for the best and worst cases. Although the worst-case runtime is no better than FastGreedy, it practically works well. They also apply the lazy evaluation trick to the double greedy algorithm, which guarantees a constant approximation ratio of unconstrained greedy maximization. Experimental results on both synthetic and real-world datasets verify the effectiveness of the proposed acceleration with a promising speedup.


**Questions:**

Some explanations of [22] are missing. The algorithm studied in [22] can be essentially used for the size-constrained greedy DPP MAP. Although it is required a one-time preprocessing, its runtime is logarithmic in n (the size of the ground item), which is much beneficial for large n. And they reported up to x224 speedup than the FastGreedy algorithm. Hence, by simple comparison to FastGreedy (LazyFastGreedy shows x17 speedup), it is expected that the algorithm is still faster than LazyFastGreedy. Does the author have a chance to compare with it?

**Limitations:**

Mentioned in weakness

**Strengths And Weaknesses:**

Strengths:
- This paper is written well and easy to follow up on the methodology.
- This paper fills the gap of the FastGreedy algorithm by adopting the lazy evaluation trick, and report the empirical speedup beyond the other algorithms for greedy submodular maximization.

Weaknesses
- The novelty of this paper is fairly weak. The main contribution is a simple adaptation of the lazy evaluation to the greedy DPP MAP inference algorithms. Both the lazy evaluation trick and the fast greedy DPP MAP implementation (as well as the double greedy algorithm) have been studied prior. These contributions are under the bar of acceptance.
- Although the proposed algorithm empirically works faster than other competitors, it is suspicious when it can be effective. Currently, the runtime complexity is lightly analyzed including only the best and worst scenarios. And the worst-case runtime is even worse than the previous FastGreedy algorithm (in the general case). This makes contributions of this work weak and it needs more careful analysis of the runtime, e.g., conditions that the LazyFastGreedy algorithm achieves for the best (or at least better) runtime.

---

> ### Author Response · Authors · 2022-08-02
> **Response to Reviewer f4YC**
>
> We appreciate the reviewer's valuable comments.
>
> &nbsp;
> ## Response to the question
> > Some explanations of [22] are missing. [...] Does the author have a chance to compare with it?
>
> Thank you for pointing it out. We will detail the difference from (Han and Gillenwater, 2020)[22] in the final version.
>
> Please note that [22] and our work are so different that they do not admit simple comparisons. First, we summarize the difference in problem settings using the notation of [22].
>
> ### Difference in problem settings
> [22] assumes customized DPPs: given a **fixed** matrix $B\in\mathbb{R}^{D\times N}$ of feature vectors, DPP instances with $L=(WB)^\top(WB)$ arrive, where only the customization matrix $W\in\mathbb{R}^{D\times D}$ can change. By contrast, we consider general DPPs with $L = B^\top B$, where $B$ changes over instances. Our algorithm is also applicable to positive semi-definite $L$ that is not decomposed as $B^\top B$, which we call the $L$-input setting in Section 5.1. Such cases naturally arise if $L_{ij}$ is given by some kernel function $\kappa(x_i,x_j)$ of feature vectors $x_i,x_j$ (see, e.g., [[Sharma et al. (2015)]](http://proceedings.mlr.press/v37/sharma15.html)).
>
> Under the customized setting, [22] pre-processes the fixed $B$ to run an (inexact) greedy algorithm in $O(kKD^2\log N)$ time, where $k$ is a hyperparameter of $k$-means clustering on $N$ data points and $K$ specifies the cardinality constraint. For general DPPs, however, our LazyFastGreedy is better regarding both (1) total runtime and (2) theoretical guarantee for the following reasons.
>
> ### 1. Total runtime
> In the general setting, the pre-processing must be done every time a new instance is given, which takes $O(kDN\log N)$ time ([22, Theorem 4]). Since LazyFastGreedy runs in $O(KN(D+\log N))$ time, ours is faster if $K < k$; this is often the case since experiments in [22] use $K=10$ and $k=50,100,500$. Moreover, in practice, [22, Figure 1 and Table 1] show that the pre-processing takes much longer than FastGreedy [13]; here, note that **the x224 speedup reported in [22] does not include the pre-processing time**. Meanwhile, our LazyFastGreedy is up to x17 times faster than FastGreedy [13]. Thus, ours is empirically faster than [22] applied to general DPPs.
>
> ### 2. Theoretical guarantee
> The algorithm in [22] aims to **approximate** solutions of the greedy algorithm (or greedy solutions), as mentioned in [22, page 3]. From [22, Theorems 2 and 3], how well it approximates greedy solutions depends on $W$. Thus, for general DPPs where $L$ may not be given via customization, the algorithm of [22] does not enjoy theoretical guarantees. By contrast, our LazyFastGreedy computes **exact** greedy solutions, thus preserving the celebrated $(1-1/\rm{e})$-approximation.
>
> To conclude, whether [22] or ours is better depends on the situation: customized or general. Since the subject of our study is general DPPs, we omitted detailed comparisons for simplicity.
>
> &nbsp;
> ## Responses to other comments
> > They also apply the lazy evaluation trick to the double greedy algorithm [...]
>
> We would like to resolve a possible misunderstanding. Our contribution to DoubleGreedy is to develop its "fast" version (not a "lazy" version). Unlike the lazy evaluation, the "fast" technique reduces the time complexity. This theoretical contribution seems to be missed.
>
> &nbsp;
> > The main contribution is a simple adaptation of the lazy evaluation to the greedy [...]
>
> Although our work is built on LazyGreedy [36] and FastGreedy [13], combining them is not straightforward, as explained in Section 3.1. Indeed, [Reviewer LKp4](https://openreview.net/forum?id=EWyhkNNKsd&noteId=sUtG3fiuyDM) says **"However, since this combination is not straightforward, this paper is interesting for the ML/DPP community."** Moreover, the fact that Chen et al. [13] rather compared FastGreedy and LazyGreedy in the experiments than combined them implies that the combination is non-trivial.
>
> &nbsp;
> > it needs more careful analysis of the runtime, e.g., conditions that the LazyFastGreedy algorithm achieves for the best (or at least better) runtime.
>
> It turned out that further analysis of the runtime was too involved. Instead, we present example situations where LazyFastGreedy runs fast.
> 1. If $d\ge\log n$ (which is often true), LazyFastGreedy achieves the same $O(knd)$ time as FastGreedy [13].
> 2. In practice, LazyFastGreedy hardly takes longer than [13] owing to the lazy update. Moreover, even if the lazy update works poorly, it can run faster due to the cache efficiency, as explained in Section 5.1.
> 3. LazyFastGreedy is expected to achieve the best runtime if $L$ is close to being diagonally dominant due to the following reason. LazyFastGreedy first sorts the elements by pushing them into the priority queue. If $L$ is diagonal, choosing the top-$k$ from the sorted elements yields a greedy solution. Consequently, it performs no redundant update and thus achieves the best runtime as in Theorem 3.2.

---

> > ### Comment · Reviewer_f4YC · 2022-08-09
> > **Comment**
> >
> > Thanks to the authors for addressing my remarks and resolving misunderstanding. I agree that the problem setting in [22] is different in this work and under the L-input setting (full-rank) LazyFastGreedy can be beneficial. Notice that the customized DPP is a general case of DPP (by defining $W$ as the identity matrix and $B$ as the Cholesky decomposition of $L$) hence it is possible to compare it to LazyFastGreedy.
> >
> > Regarding the third response, it is interesting that LazyFastGreedy with a diagonally dominant matrix can be more effective. I suspect that this can be related to the spectral property of L and if more quantitative analysis, e.g., a correlation between the number of calls for UPDATEROW subroutine is given, the contribution is much stronger. However, the current result is not sufficient even though LazyFastGreedy practically runs faster than other greedy implementations.
> >
> > I believe that the contributions are still incremental and there is headroom for improvement on the runtime analysis. Hence, I keep my current rating.

---

> > > ### Author Response · Authors · 2022-08-10
> > > **Thanks to the comment**
> > >
> > > We appreciate the reviewer's additional detailed comments.
> > >
> > > We would like to clarify what if we compare LazyFastGreedy with [22] by regarding $W$ as an identity matrix.
> > > Experiments in [22] suggest that if we do so, the method of [22] takes longer than FastGreedy (and hence slower than LazyFastGreedy).
> > > This is because $W = I$ has no information about new instances; thus, $B$ must be pre-processed every time.
> > >
> > > Also, thanks to the valuable suggestion about the relation between the spectral property of $L$ and the time complexity.
> > > Currently, we do not have a clear answer about the relationship. We will mention it as an important future direction.

---

### Official Review · Reviewer_RhmJ · 2022-07-11

**Rating:** 7
**Confidence:** 3
**Soundness:** 4 excellent
**Presentation:** 3 good
**Contribution:** 3 good

**Summary:**

This paper presents new greedy algorithms for maximum a posteriori (MAP) inference for determinantal point processes (DPPs).  MAP inference seeks to find the subset with the highest probability under the DPP.  The first major algorithm presented by the authors combines the ideas of “lazy” and “fast” greedy MAP inference algorithms from prior work.  This LazyFastGreedy algorithm has nearly the same time complexity as the fastest known FastGreedy algorithm from prior work, but runs faster in practice; it is applied to cardinality-constrained DPP MAP inference and extended to several other variants of greedy algorithms.  The second major algorithm presented by the authors is a fast version of the DoubleGreedy algorithm for unconstrained DPP MAP inference.  Finally, the authors present experimental results on synthetic and real datasets, demonstrating that their proposed algorithms can be significantly faster than the fastest known approaches from prior work.

**Questions:**

* On line 220, d >= n seems to be incorrect.  I suppose this should instead be d <= n.
* As indicated above, Figure 1(b) should be revised to be more clear.
* As indicated above, can you describe how the item vectors of the B matrix for the real-world datasets are computed?
* Nonsymmetric DPPs (NDPPs) have recently gained some attention.  Are the MAP inference algorithms presented in this paper appropriate for NDPPs?


**Limitations:**

A brief mention of the limitation of the LastFastGreedy algorithm not improving upon the worst-case time complexity of the greedy algorithm is included in the paper.  However, the authors should provide a somewhat more detailed discussion of any other known limitations, as well as the potential for negative societal impact, which is completely absent.

**Strengths And Weaknesses:**

Strengths:
* The new greedy DPP MAP inference algorithms presented by the authors appear sound, and are reasonably well explained.
* The derived runtimes appear correct, although I have not carefully validated the proofs.
* The experimental results do indeed demonstrate that the proposed algorithms can be significantly faster than the fastest known approaches from prior work, despite nearly equivalent time complexity (in the case of LazyFastGreedy).  Therefore, the contributions in this paper are significant.
* The discussion of related work is reasonably clear and thorough.

Weaknesses:
* For experiments on the real-world datasets (Netflix Prize and MovieLens), the authors don’t explain how the item vectors of the DPP B matrix are computed from each dataset.  This is an important missing detail that should be provided, since each of these datasets consists of users’ ratings of items (movies), and it is not obvious how each item vector was built from these ratings.
* Figure 1(b) is somewhat unclear.  This figure should be reworked to be more clear and interpretable.

---

> ### Author Response · Authors · 2022-08-02
> **Response to Reviewer RhmJ**
>
> We are grateful to the reviewer for providing many constructive comments.
>
> &nbsp;
> > On line 220, d >= n seems to be incorrect. I suppose this should instead be d <= n.
>
> It may appear counterintuitive, but $d \ge n$ is correct. This is because we here consider the unconstrained setting, which requires $L = B^\top B$ to be non-singular, where $B \in \mathbb{R}^{d \times n}$. Otherwise, $\det L = 0$ and thus $f(S) = \log \det L[S] = -\infty$ for $S = [n]$. Note that $f(S) > -\infty$ is commonly assumed in this context.
>
> &nbsp;
> > As indicated above, Figure 1(b) should be revised to be more clear.
>
> Thank you for the valuable suggestion. We will do our best to make a more clear figure. (We would appreciate it if you could kindly tell us why you found it unclear.)
>
> &nbsp;
> > As indicated above, can you describe how the item vectors of the B matrix for the real-world datasets are computed?
>
> Thank you for the comment. We below present the detail of how to compute $B$. In the final version, we will add it to the current description in lines 270--272.
>
> The way of computing $B$ is identical to that of [13]. Given a rating score $s \in \\{1, 2, 3, 4, 5\\}$, we binarize it by setting it to $1$ if $s \ge 4$ and $0$ otherwise. By applying this binarization to the $d \times n$ rating matrix in the datasets, we obtain $B \in \\{0, 1\\}^{d \times n}$. If the resulting $B$ has all-zero rows or columns, we remove them since they are redundant. In this way, we computed $B$ from the datasets.
>
> &nbsp;
> > Nonsymmetric DPPs (NDPPs) have recently gained some attention. Are the MAP inference algorithms presented in this paper appropriate for NDPPs?
>
> Thank you for the interesting question. We checked [Gartrell et al. ICLR 2021](https://openreview.net/forum?id=HajQFbx_yB) and examined whether our algorithm is extendable to nonsymmetric DPPs (NDPPs). Unfortunately, we found that we could not immediately extend it to NDPPs. The main difficulty comes from the fact that the log-determinant function of NDPPs lacks submodularity. In (Gartrell et al. ICLR 2021, Theorem 2), the authors obtained an approximation guarantee by using a notion of an approximate submodularity (i.e., the submodularity ratio). In our case, however, we need exact submodularity for the lazy update to work correctly. We will mention the extension to NDPPs as a future research direction.

---

> > ### Comment · Reviewer_RhmJ · 2022-08-08
> > **Rebuttal feedback**
> >
> > I thank the authors for the comments in their rebuttal.  After reading their rebuttal comments, as well as the other reviews and rebuttal comments, my rating remains a 7 (accept).  I believe that the authors have adequately addressed concerns regarding novelty and significance of the contributions raised by reviewer f4YC.
> >
> > Regarding Figure 1, it would be helpful for the authors to include more explanation of this figure.  For example, when are the marginal available in the LastFastGreedy algorithm in Figure 1(b)?  Also, what does the arrow pointing to the right, at the box of this figure, represent?

---

> > > ### Author Response · Authors · 2022-08-09
> > > **Thanks to additional feedback**
> > >
> > > We appreciate the valuable feedback on Figure 1. It greatly helps us to improve its clarity. We would like to answer the questions on the figure.
> > >
> > >
> > > >when are the marginal available in the LastFastGreedy algorithm in Figure 1(b)?
> > >
> > > An exact marginal gain (or an exact diagonal entry) becomes available once all off-diagonals in the same row are computed (i.e., when the shaded area reaches the diagonal).
> > > The diagonal entries, however, always maintain upper bounds on marginal gains even if some off-diagonals are not computed, thus enabling the lazy update.
> > >
> > >
> > > >what does the arrow pointing to the right, at the box of this figure, represent?
> > >
> > > The arrow indicates the direction in which the computation of off-diagonals proceeds.
> > >
> > >
> > > We also thank the reviewer for carefully checking our responses to the other reviewers and providing positive comments. Please feel free to ask any questions if whatever minor concerns remain.

---

### Official Review · Reviewer_tSLD · 2022-07-23

**Rating:** 5
**Confidence:** 4
**Soundness:** 3 good
**Presentation:** 3 good
**Contribution:** 2 fair

**Summary:**

This paper proposes the acceleration implementations of the greedy-type algorithms for DPP MAP inference. Two algorithms are developed for the cardinality-constrained and unconstrained DPP MAP inference respectively.  The computational efficiency and accuracy are validated through experiments on simulated and real-world datasets.

**Questions:**

1.	The authors propose two algorithms for the two different tests. What is the motivation to propose these two types?
2.	The introduction section of the relevant background is unclear and needs more details and formalization.
3.	In section 5.1, figure 2, we observe the runtime is not strictly increased when $n$ increases. Why do the authors get decreasing runtime values?
4.	In section 5.2, please provide more synthetic data settings when $n$ and $k$ vary.
5.	I suggest the authors demonstrate the experiment results in more detail and show the power of the purposed methods.
6.	The authors review many existing methods in the introduction but only compare three methods in the simulation section. Why do you choose these three methods to make a comparison?


**Limitations:**

The authors mention the limitation that the method does not improve the worst-case time complexity. The proposed method can only fasten empirical runtimes.

**Strengths And Weaknesses:**

Originality: This paper incorporates the ideas of “lazy” and “fast” which are thought to be incompatible in the previous literature.
Quality: The experiment results show that the algorithms are powerful.  The paper is generally well written in both aspects of grammar and technical discussion.
Clarity: A lot of details of the experiment seem to be missing.
Significance: The proposed acceleration method does have effectiveness through validations on simulated and real datasets.

---

> ### Author Response · Authors · 2022-08-02
> **Response to Reviewer tSLD**
>
> We thank the reviewer for providing thoughtful comments and clearly listing the questions. We below respond to them one by one. We would appreciate your possible consideration of raising the score if the questions are resolved.
>
> &nbsp;
> > 1. The authors propose two algorithms for the two different tests. What is the motivation to propose these two types?
>
> This is because the two problem settings, cardinality-constrained and unconstrained, are fundamentally different. Both are important in the context of diverse selection (e.g., [13, 20]), and the choice of algorithms depends heavily on the settings: in the cardinality-constrained case, algorithms like the standard greedy [40] for *monotone* submodular maximization are effective, while algorithms for *non-monotone* submodular maximization, such as the DoubleGreedy [11], is usually used in the unconstrained case. We studied the two algorithms separately to cover the two important and different settings.
>
> &nbsp;
> > 2. The introduction section of the relevant background is unclear and needs more details and formalization.
>
> Thank you for the suggestion. We will do our best to make the introduction clear by presenting more details of the background.
>
> &nbsp;
> > 3. In section 5.1, figure 2, we observe the runtime is not strictly increased when $n$ increases. Why do the authors get decreasing runtime values?
>
> First, we would like to fix an error in Figure 2 (f): the y-axis label must be "Computed off-diagonals," not "Runtime."
>
> Generally speaking, the computation cost of our LazyFastGreedy tends to increase as $n$ grows, but larger $n$ does not always mean strictly longer running times in practice. This is because the running time is affected by how well the lazy update works, which depends on the entries of $L$. Thus, even if $n$ gets larger, the running time can sometimes decrease owing to the lazy update.
>
> &nbsp;
> > 4. In section 5.2, please provide more synthetic data settings when $n$ and $k$ vary.
>
> We conducted additional experiments. Below is the running time [sec] of DoubleGreedy and FastDoubleGreedy with various $n$ values. Similar to the case with $n = 6000$ shown in the paper, our FastDoubleGreedy is much faster than the original DoubleGreedy for every $n$. Please note that the problem in Section 5.2 is unconstrained; thus, the parameter $k$ to specify the cardinality constraint does not appear here.
>
> #### **DoubleGreedy (existing)**
> |   $n$   | Product |   Greedy   | **Total** |
> | ------: | ------: | ---------: | --------: |
> |  $2000$ |   $1.1$ |    $258.6$ |   $259.7$ |
> |  $4000$ |   $8.7$ |   $5422.4$ |  $5431.1$ |
> |  $6000$ |  $30.0$ |  $36233.1$ | $36263.1$ |
> |  $8000$ |  $70.2$ | $>86400.0$ |    ----   |
> | $10000$ | $137.9$ | $>86400.0$ |    ----   |
>
> #### **FastDoubleGreedy (proposed)**
> |   $n$   | Product | Inverse |  Greedy |**Total**|
> | ------: | ------: | ------: | ------: | ------: |
> |  $2000$ |   $1.1$ |   $1.8$ |   $0.7$ |   $3.6$ |
> |  $4000$ |   $8.7$ |  $16.9$ |  $10.0$ |  $35.6$ |
> |  $6000$ |  $30.0$ |  $59.8$ |  $34.6$ | $124.4$ |
> |  $8000$ |  $70.2$ | $151.2$ | $103.1$ | $324.5$ |
> | $10000$ | $137.9$ | $294.2$ | $182.6$ | $614.7$ |
>
> &nbsp;
> > 5. I suggest the authors demonstrate the experiment results in more detail and show the power of the purposed methods.
>
> Thank you for the suggestion. We will elaborate more on the experiments by, e.g., presenting improvement ratios using an additional page in the camera-ready version if accepted.
>
> &nbsp;
> > 6. The authors review many existing methods in the introduction but only compare three methods in the simulation section. Why do you choose these three methods to make a comparison?
>
> We used the three methods, Naive [40], Lazy [36], and Fast [13], in Section 5.1 since they are particularly relevant to our goal, fast computation of greedy solutions.
> Also, as for [10], we implemented a Fast version and conducted experiments in Section 5.2.
> In addition, regarding [10, 27, 37, 42], we implemented Lazy and/or Fast versions and conducted experiments in Appendix A.4. The other methods are not experimentally compared for the following reasons:
>
> - [23] has already been improved by Fast [13].
> - [22] considers a different problem setting than ours. For details, please see the [response to Reviewer f4YC](https://openreview.net/forum?id=EWyhkNNKsd&noteId=MGJ__6kKSqn).
> - [20] studies a continuous relaxation method, which is well known to be much slower than combinatorial greedy-style methods like ours.
> - [15, 34, 35, 6, 41] study a different problem setting of the determinant maximization (without log).
> - [2, 32, 16, 21, 1, 12, 31] study sampling methods, not MAP inference methods.
> - [19, 40] are fundamental results of submodular maximization and are not concerned about fast algorithms.
> - [3, 4, 18, 8, 17, 14, 28] assume a different computational environment than ours. They study parallel algorithms assuming polynomially many processors are available.

---

### Meta-Review · Area_Chair_6WUm · 2022-08-27

**Recommendation:** Accept
**Confidence:** Certain

**Metareview:**

Although the reviewers gave a wide range of ratings to this paper, they all agreed that it is well-written and presents two novel, sound algorithms that perform well in practice. The main concern was that the algorithms merely combine existing techniques, which is true. However, given that the resulting algorithms are elegant, well-motivated, and highly performant, and that achieving the combinations is not trivial, I believe this paper makes a significant contribution to an active field of research and deserves acceptance.

The authors should take the reviewers' comments into account as they prepare their final revision. In particular, I would encourage them to more explicitly describe their reasons for presenting two different algorithms (cardinality constrained/unconstrained). Currently, it feels like the two parts of the paper are somewhat disconnected.

**Award:**

No

---

### Decision · Program_Chairs · 2022-09-14

Accept